# Health Literacy in Early Childhood: A Systematic Review of Empirical Studies

**DOI:** 10.3390/children9081131

**Published:** 2022-07-28

**Authors:** Henrietta Bánfai-Csonka, József Betlehem, Krisztina Deutsch, Martina Derzsi-Horváth, Bálint Bánfai, Judit Fináncz, Judit Podráczky, Melinda Csima

**Affiliations:** 1Institute of Emergency Care and Pedagogy of Health, Faculty of Health Sciences, University of Pécs, 7621 Pécs, Hungary; jozsef.betlehem@etk.pte.hu (J.B.); krisztina.deutsch@etk.pte.hu (K.D.); balint.banfai@etk.pte.hu (B.B.); 2Doctoral School of Health Sciences, Faculty of Health Sciences, University of Pécs, 7621 Pécs, Hungary; horvath.martina06@gmail.com; 3Department of Emergency Medicine, Medical School, University of Pécs, 7624 Pécs, Hungary; 4Institute of Education, Hungarian University of Agriculture and Life Science, 7400 Kaposvár, Hungary; financz.judit@uni-mate.hu (J.F.); podraczky.judit@uni-mate.hu (J.P.); petone.csima.melinda@uni-mate.hu (M.C.)

**Keywords:** early childhood, health literacy, kindergarten, preschool children

## Abstract

Early childhood plays a key role in the formation of healthy habits and the establishment of health literacy. Nonetheless, there are only a few research studies focusing on the health literacy level of children under the age of eight. The aim of our systematic review is to explore empirical research on health literacy related to early childhood. The research was conducted in accordance with the PRISMA protocol. This systematic review examines 12 studies published between 2013–2022. Results show that research focuses on different domains of health literacy for children. In relation to children’s food literacy, children understand the relationship between health and nutrition and they realize the health impact of obesity. The habits connected to oral health are strongly associated with parents’ knowledge of and behaviours around oral health. Results related to health care situations show that children are able to be actively involved in decision-making processes in connection with their health. Exploring young children’s health literacy is essential in order to be able to plan health promotion interventions, embedded into early childhood education. Picture-based messages or story-based messages supported by illustrations can help measure health literacy in early childhood and can support the formation of health literacy.

## 1. Introduction

The concept of health literacy (HL), after its appearance in 1974, was approached primarily from a biomedical perspective in the 1980s and 1990s, but its interpretation has since become multidisciplinary [1]. Health literacy is a growing research area, but only a small amount of literature deals with HL among young children. Health literacy in childhood is fundamental to physical, emotional and cognitive development and the evolvement of health-related behaviours [2]. In postmodern society, issues about health and healthy lifestyle have come into focus, along with the skills and competencies that enable health promotion, which have also become more valuable. In this context, there is an increasing emphasis on prevention, to which special attention should be paid in early childhood [3,4,5]. Greater effectiveness is expected from health promotion activities during early childhood instead of changing the minds of adults. In addition to the family, institutions, caregivers and health care professionals play a role in establishing an appropriate level of health literacy and healthy behaviour [6]. Many studies deal with the health literacy of parents, caregivers and professionals working in health care and in early childhood education [7,8,9,10,11,12,13]. Studies on teachers’ health literacy has focused on two domains of the Sorensen typology [14], that is, their knowledge and competencies related to prevention and health promotion [15,16], whereby mental health literacy appears as a separate area [8,17,18].

Most research focusing on parents’ health literacy confirms the relationship between the HL level of parents and their children’s health outcomes [19,20,21]. In addition, to measure general HL among parents, several studies examined oral- [22,23], fever- [24,25], and pharmacotherapy literacy [26,27]. Disadvantaged parents with lower social status are emphasized in the design of studies on parental health literacy [28,29]. Conclusions of these studies draw attention to the positive changes in the health status and health behaviours of children through the formation of parental health literacy. This is especially important in the case of deprived families, to reduce inequalities in health.

Although there has been an increase in research interest in children’s health literacy in recent years, there are still very few studies available for this age group [2]. The reason for this is primarily due to the lack of measuring instruments suitable for examining their health literacy. To fill this gap, the children’s version of the European Health Literacy Survey Questionnaire was developed and tested in Germany (HLS-Child-Q15), which is suitable for measuring the health literacy of children over eight years old [30,31].

General health literacy is also an interesting area in early childhood health literacy education. Measuring children who cannot write and read is not an easy task. General health literacy is not a specific area; later it can help children to navigate the health care system. Some measurements have different indexes of health literacy, such as health care, health promotion and disease prevention [14]. A potential topic for future research is to identify possible ways to obtain a complex view of health literacy in early childhood from a general point of view.

HL among children is a complex question because according to the European Health Literacy Consortium, health literacy is “linked to literacy and entails people’s knowledge, motivation and competences to access, understand, appraise, and apply health information in order to make judgments and take decisions in everyday life concerning health care, disease prevention and health promotion to maintain or improve quality of life during the life course” [14] (p. 3). However, children in most countries do not have the adequate independence and ability to make their own decisions about their own health, since parents are authorized to decide for them [32]. For children younger than 12, two definitions are used: (1) “The meaning of health literacy to children is defined as “to understand and act upon physical and psycho-social activities with appropriate standards, being able to interact with people and cope with necessary changes and; demands reasonable autonomy so as to achieve complete physical, mental and social well-being”” [33] (p. 257); (2) “[...] health literacy was defined simply as the ability to understand health information and to understand that actions taken in youth affect health later in life, combined with the ability to access valid health information” [34] (p. 13). To measure HL level in childhood there are a lot of instruments that focus on different areas [35,36,37,38], but we could not find even one which focused on general HL level in early childhood.

In international terminology, there is no consensual definition regarding early childhood. It depends on local traditions and the organization of the school system. Early childhood is the period of life under eight years of age, according to the UN Convention on the Rights of the Child, which considered and took into account the differences among the various developmental theories [39].

Despite the fact that in recent years, examining the effects of interventions during early childhood has become a priority for researchers [40,41], only a few studies deal with the HL level of children under the age of eight [42,43,44,45,46,47,48,49,50,51,52,53]. However, studying this field is essential because the establishment of a preventive health attitude and the formation of health-related good habits occur in early childhood [15].

In order to reduce the perceived shortage in this area of research, the aim of our present study is to systematically analyse empirical research on health literacy related to children under the age of eight. The following research questions guided this systematic review according to the PICO [54] format:What knowledge is available regarding the health literacy of children under eight?In which areas of health literacy has research been conducted among young children?What are the issues that require further exploration?What kind of methods are used to explore health literacy in early childhood?What are the achieved outcomes of applied interventions related to the health literacy of young children?

Answering these questions can contribute to supporting pedagogical activities regarding the content of health literacy and health promotion methodology, which can be incorporated into early childhood education and intervention.

## 2. Materials and Methods

Our research was conducted according to the Preferred Reporting Items for Systematic Reviews and Meta-Analyses (PRISMA) 2020 protocol [55]. The protocol outlining the aims and scope of this systematic review was registered with PROSPERO (number: CRD42022343699) [56].

### 2.1. Eligibility Criteria

Eligibility criteria were based on the PICOS (Participants, Intervention, Comparison, Outcomes, Study design) framework. Studies were included if they complied with the following criteria: (1) written in English language; (2) peer-reviewed; (3) focused on health literacy in early childhood; (4 empirical studies (such as cross-sectional quantitative and qualitative studies, e.g., survey, observation, picture-based studies, as well as baseline data from intervention studies); (5) participants aged under eight years. Studies were included if they were considered an empirical implementation study (i.e., original data collection) and statistically tested or qualitatively explored mechanisms, mediators, or moderators.

Articles were excluded if (1) the full text was not available; (2) types of papers were any of the following: secondary analysis, content analysis, document analysis, systematic and other reviews, or validation of measuring tools without any results.

### 2.2. Information Sources

The research sample was defined in Scopus, WOS, ERIC, and PubMed databases by specifying the keywords “early childhood” or “preschool” or kindergarten and “health literacy”. We did not use any other sources to find literature. The combination of these words was searched in title, abstract and keywords. We interpreted the “early childhood” stage of life according to the UN Convention on the Rights of the Child, being from 0 to 8 years of age [39]. The period of early childhood includes the preschool and kindergarten stage; however, primary or elementary school goes beyond this period (6/7–10/12 year old children, depending on the country’s school system). 

As an additional filter, we determined that the publication year of the examined studies should be between 2013 and 2022. 

### 2.3. Selection Process

In the first phase of the search, the WOS database identified 114 results, Scopus 426, PubMed 345 and ERIC 9 results. Out of a total of 894 results, duplicates were removed in the first step; a total of 381. After reviewing the titles and abstracts (MCs) of the remaining 513 records, we excluded from the analysis those in which the age of the children and the content of texts did not meet the selection criteria. Regarding age, it should be noted that in many papers the age of the children included in the research varied quite widely. Although early childhood fell within the sample age range, the researchers did not report separate results for early childhood, so these studies were excluded from further analysis. We also excluded systematic and other reviews, studies that did not include empirical research, and those that did not appear in a journal. We assessed 238 full articles (MCs, HBCs, JF), 18 of which proved to be relevant to the eligibility criteria of the study topic. An additional 6 articles were excluded due to specific reasons. Following the inclusion and exclusion criteria, 12 articles were selected for the systematic review (Figure 1). Disagreements which emerged during the selection process were resolved through discussion with members of the research team.

All references were imported into a Microsoft Excel table and duplicated reports were removed before the screening. At each major step of this systematic review, discrepancies between authors were resolved through discussion.

### 2.4. Quality Assessment

For critical appraisal and to limit bias, each included study underwent critical review. For cross-sectional quantitative studies, we used the Newcastle–Ottawa cohort scale to assess for bias [57] (Appendix A). According to this points system, three main criteria are evaluated: selection, with a maximum of 5 stars; comparability, with a maximum of 2 stars; and outcome, with a maximum of 3 stars.

For cross-sectional qualitative studies, we used a special tool [58], the results of which are described in Appendix A. The scale is a 4-point Likert scale where 1 point means “very poor”, 2 points means “poor”, 3 points means “fair”, and 4 points means “good”. The quality of the studies can be defined as high (A), medium (B) or low (C). High-quality studies have points between 30–36, medium-quality studies have points between 24–29, and low-quality studies have points under 24.

Cochrane Risk of Bias criteria were used to explore the risk of bias in the interventional studies [59] (Appendix A). According to this evaluation system the research is evaluated according to seven elements (random sequence generation (selection bias), allocation concealment (selection bias), blinding of participants and personnel (performance bias), blinding of outcome assessment (detection bias), incomplete outcome data (attrition bias), selective reporting (reporting bias) and other sources of bias). It is not a points system, rather, a judgement of quality, categorized as: high risk (red), unclear risk (yellow) and low risk (green). 

The results of the quality assessment can be found in Appendix A.

## 3. Results

During our first search using the keywords, we found 894 studies. After removing the duplicates, 513 studies remained. The 12 studies reviewed in the analysis present different domains of health literacy pertaining to children [42,43,44,45,46,47,48,49,50,51,52,53], which deal with different areas of health literacy among children under eight years old (Table 1).

### 3.1. Characteristics of Included Studies

The general characteristics of the included studies are summarized in Table 1. Eight of the studies were published in the last four years (2018–2022) [42,44,45,49,50,51,52,53]. Seven of them were conducted in Europe (Table 1) [44,45,48,49,50,51,52]. Four studies contained interventions among children [42,43,50,53] (Appendix A).

During the analysis of the studies we found that most of them were good quality studies and had a high-evidence research level. Tabacchi et al. [51] authored the best quality cross-sectional quantitative study from the selected 12 studies [42,43,44,45,46,47,48,49,50,51,52,53]. Among the qualitative research studies, we did not find any that rated worse than medium quality [44,45,46,47,48]. During the evaluation of the studies, we found a number of studies where the risk of bias was high or unclear. Despite this fact, we thought that these studies could contribute to answering our research questions and help us to obtain a wider picture about health literacy levels in early childhood.

Two studies measured the HL level among children aged 5–11 [46] and 5–10 [47], but the results were presented separately for each different age group so they were not excluded from the review. Most of the studies included children aged 3–6 years [42,43,44,45,48,49,50,51,52,53] (Appendix A).

### 3.2. Health Literacy in Early Childhood

General health literacy is difficult to measure at a young age; however, three studies focused on this issue in health care settings or in the field of health promotion [43,45,48]. Separate areas such as oral health literacy [42,43,53] and food health literacy [44,45,46,47,49,50,51] appeared more frequently; in addition, one study focused on the specific area of stroke health literacy [52].

#### 3.2.1. Health Care

Stalberg et al. (2016) explored children’s perceptions of being in a health care situation. According to their results, children can be actively involved in decision-making processes, and they demand direct information from professionals working in health care [48]. Only one study focused on children’s knowledge of the symptoms of a direct disease, but it contributes to finding out what competencies they have about stroke and how we can teach this knowledge to children. Children can recognize such time-dependent situations as stroke, but some children at this age do not have sufficient knowledge about how to react appropriately in the event of a stroke [52].

#### 3.2.2. Food Literacy

Most of the studies examined children’s food literacy [44,45,46,47,49,50,51]. They explored the children’s orientation in terms of connection to healthy or unhealthy nutrition. Related to this, they focused on the assessment of obesity [44], the type of foods that children choose as healthy food [47], and nutrition in children’s perception [45,46,49,50,51]. They used picture-based tests administered through face-to-face measurements. Special attention was paid to children’s healthy dietary choices. According to these studies, children had an overall good knowledge of the main food categories, they could recognize the healthy and non-healthy foods and understand the relationship between portion and health and between food and health; furthermore, they could speak about the health impact of obesity.

#### 3.2.3. Oral Health Literacy

Oral health literacy is strongly associated with parents’ knowledge and behaviour about oral health. Parents with a higher HL level can more effectively help their children to act healthier and have better practices in oral health [43]. In the SIMS program teachers and parents can help children to form good brushing habits [42]. For developing healthy habits in early childhood, stories and pictures can also be useful [53].

#### 3.2.4. Health Promotion

Although only one study directly examined children’s experiences and attitudes towards health [45], others also mentioned children’s perceptions about health competencies related to health promotion [44,51]. Children at the age of four are health-conscious and can recognize basic health concepts, and they are able to take an active role in their health [45].

## 4. Discussion

In our systematic review, we focused on research that dealt with measuring health literacy level in early childhood. In recent decades, many empirical studies have been conducted, but they are focused on the HL level of parents, caregivers, and teachers [7,8,9,10,11,12,13], and conclude their effects about children’s health outcomes and health behaviour [19,20,21]. We found only 12 research studies that focused directly on HL level among children [42,43,44,45,46,47,48,49,50,51,52,53].

The main approaches of studies dealing with HL in early childhood are as follows: (1) investigating the parents’ HL level to conclude the child’s health and health behaviour; (2) qualitative methods are mostly used; (3) measuring knowledge and skills; (4) the main study questions focus on special areas, such as food and nutrition, oral health and health care situations. Health literacy is a popular research area, and many study protocols, tools and validation studies related to young children have been published in recent years [60,61,62]. We can absolutely agree with these types of approaches, because children at this young age do not have the ability to write and read, and qualitative methods are the best to measure their abilities.

Studies included in the systematic review were separated into two groups. The first group of articles did not have any intervention; they were mostly cross-sectional studies [44,45,46,47,48,49,51,52,53]. Because of the age characteristics of young children, HL in early childhood can be measured indirectly, through the development of good health-related habits. Researchers consider it important to find appropriate methods to measure HL level among young children. Given the lack of emergent literacy, it is very challenging to choose the right methods. According to the reviewed articles, picture- or story-based, face-to-face studies prove to be the most effective [44,45,46,48,49,52,53]. In some studies, researchers used a projective test where they asked children to tell their thoughts about a picture [45,46,48]. In other studies, children were asked to choose a picture that matched the story [44,49,52]. These methods are popular because a picture-based or a story-supported health message can improve children’s understanding of health information and support their health literacy [48,63]. A simulated environment appeared also to be an effective solution, where the researchers focused on the children’s behaviour and choices in a simulated situation [47]. In terms of topics, the studies typically focused on areas that receive special attention in health education during early childhood, including food and nutrition, physical activity, and oral hygiene [64,65]. The development of social and emotional competencies plays an important role in health education during early childhood. This also has a positive effect on children’s mental health [66,67,68]. Notwithstanding, the mental and social health dimensions do not appear in studies related to early childhood health literacy, according to our systematic review. In addition to content related to health education, research also reveals children’s perceptions of health and their experiences of the health care system [45,48]. Researchers have also investigated issues such as children’s knowledge of stroke [52]. In addition to health knowledge, health-related skills are examined in the reviewed studies.

One of the components of health literacy refers to accessing and interpreting health-related information [14]. Although decisions about children’s health are made by their parents, we must see children as capable of making independent decisions about the issues that affect them. Studies dealing with children’s perception of health care situations confirm that children are important actors who like to be actively involved in health care situations. Furthermore, they prefer to be provided with information directly from professionals, and are able to be active and reflective in interpreting health messages. Understanding children’s perceptions enables professionals to involve children in decisions and improve the level of the children’s health literacy [48].

Nowadays, unhealthy diet, obesity and a lifestyle without physical activity are predictors of chronic diseases [69]. Recognizing the importance of a healthy diet and forming habits for a healthy lifestyle, health education and health promotion are all essential in early childhood [70]. Oral health literacy is also a fundamental topic in early childhood health literacy. In some of the studies reviewed, caregivers, teachers and parents were involved in the interventions [42,43,53]. Parents’ health literacy level and toothbrushing habits are strongly associated with children’s habits [42,43].

From the view of health promotion, research draws attention to young children’s (boys were asked in the study) review of health as being dominated by a biomedical food and nutrition discourse. They perceived the concept of health to be inextricably intertwined with food and nutrition. The role of a more holistic approach to health promotion and health-promoting behaviours is a quintessential priority in early childhood [46].

As a result of our systematic review, the following topics appeared in relation to the examination of health literacy in early childhood: nutrition, physical activity, sedentary time, oral hygiene, stroke literacy, as well as being in a health care situation. In addition to these investigated factors, more attention should be paid to the examination of other health-protecting behaviours (e.g., hand washing; activities related to body care). In addition to health-protecting habits, and skills and knowledge related to health behaviour, we cannot ignore knowledge about habits that are harmful to health. In this context, it can be stated that in the studies included in our analysis, studies related to knowledge about health-damaging habits were not even mentioned, even though in many cases children are already exposed to harm from the smoking habits of their parents and their environment before they are born. Knowledge of the harmful effects of smoking, and the willingness of the young child to be dismissive of adults smoking, protects the young child from various respiratory illnesses and other diseases that appear in adulthood, such as cardiovascular diseases. For this reason, we believe that research exploring the health literacy of children younger than eight years old should be extended to include health literacy regarding the causes of health-damaging behaviour [71]. In addition, exploring health literacy content related to mental and social health dimensions would further expand our knowledge of the health literacy of young children.

The main benefit of research on young children’s health literacy is the identification of direction for the development of health promotion and health education. Since the majority of children in this period of life do not have basic literacy skills, other methods must be used to measure their health literacy.

In the course of our research work, we did not find any systematic review that examined health literacy levels in relation to early childhood. To the best of our knowledge, our systematic review is the first in this field. In our systematic review, we explored the main topics related to health literacy in early childhood and the methodological options that might be suitable for measuring health literacy among young children. In addition, we have highlighted the neglected research areas in this field.

As a limitation of our review, we can mention that only English language empirical studies were involved. In most of the studies, not only the children but also their parents were involved, and the research drew conclusions with their results. Also, there were studies where the age of the target group was wider, and the results were not separated into age groups.

## 5. Conclusions

Early childhood health education and health promotion are primarily based on activities related to the daily routine. During the formation of habits, the emphasis is on the development of abilities and skills through activity and action. As such, research about health literacy in early childhood is based on healthy habits and competencies.

The reviewed studies demonstrate that research focusing on health literacy is relevant at this early stage of life and provides results that can be incorporated into both health science and educational practice. The cognition of young children’s health literacy, and exploring their perceptions of health and a healthy lifestyle is essential in order to be able to plan health promotion interventions embedded into early childhood education. Studies included in the analysis showed that picture-based messages or story-based messages supported by illustrations can be appropriate tools for sharing health information in early childhood. In addition to supporting the formation of health literacy, these tools are also suitable for measuring it.

Research examining the health literacy of children under the age of eight is still not outstanding, and the occurrence is quite sporadic among existing relevant research. However, research interest in this age group is turning, as indicated by the fact that there are a number of study protocols in recent literature whose results are expected to expand the knowledge of health literacy regarding young children.

## Figures and Tables

**Figure 1 children-09-01131-f001:**
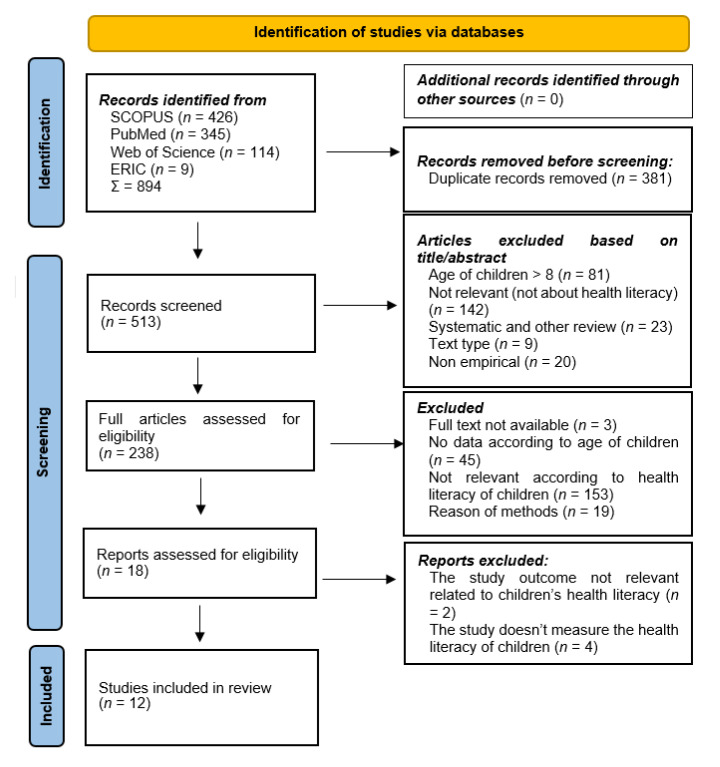
Flow chart of the searching process according to the PRISMA protocol.

**Table 1 children-09-01131-t001:** Selected empirical studies and the HL dimension analyzed.

Author/Year	Country	Title	Journal Source	General HL Dimesion *	Specific HL ** Dimension
Anwar et al. (2020) [42]	Malaysia	Effect of the SIMS program on oral hygiene levels of 5–6-year-old children in the Kampar District, Malaysia: A randomized children in the Kampar District, Malaysia: A cluster-randomized controlled trial	Makara Journal of Health Research	Disease prevention	Oral health
Brega et al. (2016) [43]	USA, Colorado	Association of parental health literacy with oral health of Navajo Nation preschoolers	Health Education Research	General HL among parents	OHL among children and parents
Charsley et al. (2018) [44]	UK	The bigger picture: young children’s perception of fatness in the context of other physical differences	Pediatric Obesity	Health promotion	Food literacy
Derwig et al. (2020) [45]	Sweden	Elucidating the child’s perspective in health promotion: Children’s experiences of child-centred health dialogue in Sweden	Health Promotion International	Health promotion	Food literacy
Drummond et al. (2013) [46]	Australia	My dad’s a ‘barbie’ man and my mum’s the cooking girl: Boys and the social construction of food and nutrition	Journal of Child Healthcare	Health promotion	Food literacy
Privitera et al. (2015) [47]	USA	Emolabeling increases healthy food choices among grade school children in a structured grocery aisle setting	Appetite	Health promotion	Food literacy
Stålberg et al. (2016) [48]	Sweden	Younger children’s (three to five years) perceptions of being in a health-care situation	Early Child Development and Care	General HL-Health care	-
Tabacchi et al. (2019) [49]	Italy	Food literacy predictors and associations with physical and emergent literacy in pre-schoolers: results from the Training-to-Health Project	Public Health Nutrition	Health promotion	Food literacy-knowledge
Tabacchi et al. (2020) [50]	Italy	Validity and Internal Consistency of the Preschool-FLAT, a New Tool for the Assessment of Food Literacy in Young Children from the Training-To-Health Project	International Journal of Environmental Research and Public Health	Health promotion	Food literacy
Tabacchi et al. (2021) [51]	Italy	An Interaction Path of Mothers’ and Preschoolers’ Food- and Physical Activity-Related Aspects in Disadvantaged Sicilian Urban Areas	International Journal of Environmental Research and Public Health	Health promotion	Food literacy
Tsakpounidou et al. (2021) [52]	Greece	Baseline Stroke Literacy of Young Children Based on “FAST 112 Heroes” Program	Frontiers in Pubic Health	Health promotion	Stroke literacy
Zhou et al. (2020) [53]	Hong Kong	Efficacy of Social Story Intervention in Training Toothbrushing SkillsAmong Special-Care Children With and Without Autism	Autism Research	Disease prevention	Oral literacy

* Sorensen’s classification [14] ** HL: health literacy.

## Data Availability

Not applicable.

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
