# Peer review of "Health Literacy in Early Childhood: A Systematic Review of Empirical Studies"

_children, 2022, doi:10.3390/children9081131_

Round 1

Reviewer 1 Report

The present SR aims in assessing health literacy in children. This is an interesting topic although several aspects need clarification. The aim of the review is very wide, not presented in PICO format. When the hypothesis tested is not presented in PICO format then it is highly likely that the scope is wide and the questions vage and unanswerable. Please adapt accordingly.

How do you answer the 5th research question (5. What are the areas where further researches are needed?)?

The protocol was not registered in any registry as detailed clearly in the PRISMA guidelines.

The search part of the methods is inadequate. How many people participated in the search, where are the exact search strings that are required in each SR as per PRISMA and Cochrane? Was grey literature included in the searches?

Where is the quality assessment of the studies?

Is there a synthesis of the findings?

What type of studies were eligible? Was this a SR of RCTs? Observational studies? This is unclear until the results! It should appear in the title.

The PRISMA flowchart is incomplete. The right part of the chart (retrieved from other sources) is missing.

Table 1 with studies characteristics is inadequate. It describes the publication of the included studies and not the studies per se. Study design, sample size, sex ratio, possible commorbidities, age range, recruitment particularities, sampling, origin, outcomes and questionnaires used, findings, these are study characteristics and must be included.

Where is the quality assessment of the studies? No SR exists without quality assessment/RoB!

Where are the limitations of the included studies?

In retrospect, it appears that this is the first attemp of the authors to perform a SR. It would be better if they re-run all the search and data exctration and include an expert in evidence-synthesis in their team. Then, they can resubmit their manuscript which will actually be a SR.

Author Response

Dear Reviewer 1,

we are very thankful for your questions, comments and suggestions to improve the quality of our paper. According to the comments we have modified our manuscript. Below you can find our answers to your questions.

1. The aim of the review is very wide, not presented in PICO format. When the hypothesis tested is not presented in PICO format then it is highly likely that the scope is wide and the questions vage and unanswerable. Please adapt accordingly.

A1: Thank you for your suggestion. We clarified the questions and rewrote in PICO format.

2. How do you answer the 5th research question (5. What are the areas where further researches are needed?)?

A2: Thank you for your question. We rewrote and restructured the research questions in the manuscript for better understanding and easier assessment.

3. The protocol was not registered in any registry as detailed clearly in the PRISMA guidelines.

A3: Thank you for your comment. Our study has been registered in PROSPERO (NUMBER: CRD42022343699).

4. The search part of the methods is inadequate. How many people participated in the search, where are the exact search strings that are required in each SR as per PRISMA and Cochrane?

A4: Thank you for your comment. We rewrote and expanded the Methods section.

5. Was grey literature included in the searches?

A5: Thank you for the question. The grey literature was also excluded.

6. Where is the quality assessment of the studies?

A6: Thank you for your comment. We added the quality assessment of the included studies in the paper. (Methods section and Table A1-A3)

7. Is there a synthesis of the findings?

A7: Thank you for the question. In the Results and Discussion sections, we tried to synthesize the main results of the included studies. We also highlighted the results with the same topics.

8. What type of studies were eligible?

A8: Following the characteristics of reviews, we included quantitative (cross-sectional, cohort, prospective, and case–control studies, as well as baseline data from intervention studies) and qualitative studies. We also mentioned it in Methods section.

9. Was this a SR of RCTs? Observational studies? This is unclear until the results! It should appear in the title.

A9: Thank you for your suggestion. In the title “systematic review” phrase appear and now we expanded it. The new title is:  “Health literacy in early childhood-a systematic review of empirical studies”

10. The PRISMA flowchart is incomplete. The right part of the chart (retrieved from other sources) is missing.           

A10: Thank you for your comment. We did not think that we need to use it, because the literature was only searched from databases and registers. The number of other searches is 0. We expanded the Methods section with this information.

11. Table 1 with studies characteristics is inadequate. It describes the publication of the included studies and not the studies per se. Study design, sample size, sex ratio, possible commorbidities, age range, recruitment particularities, sampling, origin, outcomes and questionnaires used, findings, these are study characteristics and must be included.

A11: Thank you for your comment. We renamed the table. All the characteristics that were listed we presented separately in the Table 1, Table A4 and Table A5.  

12. Where is the quality assessment of the studies? No SR exists without quality assessment/RoB! Where are the limitations of the included studies?

A12: Thank you for your comment. We added the quality assessment of the included studies in the paper. (Methods section and Table A1-A3).

We made some other corrections, clarifications and additions. Please find our corrections in the new uploaded manuscript and appendix.

We kindly request you to accept the manuscript for publication.

Yours sincerely, on behalf of the authors,

Henrietta BÁNFAI-CSONKA

subject teacher, PhD student

University of Pecs
Faculty of Health Sciences
H-7621 Pecs, Vorosmarty street 4.
Phone: +36/72/513-671
E-mail: henrietta.csonka@etk.pte.hu

Reviewer 2 Report

This paper addresses health literacy research and children under 8 years old. There is limited research on children under 8 years old in health literacy, since reading has always been a key skill in health literacy and children under 8 are not yet competent readers. However, recent arguments support claims that critical analysis of health information, verbal communication, and self-efficacy are still necessary for young children to navigate health decisions that they have to make. This paper is a systematic review of the published literature on this young population.

Unfortunately, the writing of this paper is weak and the errors throughout in grammar, sentence structure, subject-verb agreement make it difficult to assess the study components that are described in the paper. The authors should revise the paper to be written well, and then reviewers can assess the quality of the work itself with greater rigor and detail.

Some general recommended revisions in addition to the writing are below.

The introduction on page 1 confusing. It seems to bring up different issues with little evidence. Revise to clearly state what the problem is – why is it important to study children under 8 and HL?

On page 2, you seem to argue why NOT to study children so young. If you add the arguments why not to study them, and add how there are few measures focused on the young child, you will have to here add a strong argument using evidence why you are doing so. Just adding the definitions of HL for young children is not enough of a supporting claim to make in the face of the argument against studying this young population.

In Method section, there needs to be a lot more details to help reviewers assess the validity and reliability of the sampling and data analysis. For example, why these particular key words? Why not by age? Why these years selected as opposed to a different year range? What do you define as empirical studies for eligibility and why only these? You state that participants must be aged under 8, but age was not a keyword, so if you searched for early childhood, preschool or kindergarten, how did you not limit studies to 5 years old and under? Why did you not include elementary school in other words? Why do you later exclude early childhood when that was a keyword? In one section you state you reduced sample from 238 to 12 and later under results you state you reduced sample from 513 to 12. Please explain this discrepancy.

Once all the methodological details are provided, reviewers can assess the results that take up just one page here, and whether they are valid. Importantly, though, what is the “so what” of the findings? What is found that we did not already know – we already knew that oral health literacy, food, health care and health promotion might be prevalent topics, so what makes these topics unique for this age group. What uniquely and distinctly characterizes 12 articles that focus on young children?

In the discussion section, please provide the contributions of the findings to theory, to research, and to practice, and what makes the study compelling enough to publish.

Author Response

Dear Reviewer 2,

We are very thankful for your questions, comments and suggestions to improve the quality of our paper. According to the comments we have modified our manuscript. Below you can find our answers to your questions.

  1. Unfortunately, the writing of this paper is weak and the errors throughout in grammar, sentence structure, subject-verb agreement make it difficult to assess the study components that are described in the paper. The authors should revise the paper to be written well, and then reviewers can assess the quality of the work itself with greater rigor and detail.

A1: Thank you for your suggestion. The paper went through English language editing by a native English speaker.

  1. The introduction on page 1 confusing. It seems to bring up different issues with little evidence. Revise to clearly state what the problem is – why is it important to study children under 8 and HL?

A2: Thank you for your comment. We restructured and expanded the Introduction section.

  1. On page 2, you seem to argue why NOT to study children so young. If you add the arguments why not to study them, and add how there are few measures focused on the young child, you will have to here add a strong argument using evidence why you are doing so. Just adding the definitions of HL for young children is not enough of a supporting claim to make in the face of the argument against studying this young population.

A3: Thank you for your opinion. We only would have like to point out the difficulties of examining small children, but at the same time we emphasized its importance. We tried to explain it better during the correction.

  1. In Method section, there needs to be a lot more details to help reviewers assess the validity and reliability of the sampling and data analysis. For example, why these particular key words? Why not by age? Why these years selected as opposed to a different year range? You state that participants must be aged under 8, but age was not a keyword, so if you searched for early childhood, preschool or kindergarten, how did you not limit studies to 5 years old and under? Why did you not include elementary school in other words? Why do you later exclude early childhood when that was a keyword?

A4: We interpreted "early childhood" stage of life according to the UN Convention on the Rights of the child from 0 to 8 years of age. International journals, organizations (e.g. EECERA) and education policy of  the EU also  consider “early childhood from 0-8 years old. The period of early childhood includes preschool and kindergarten stage, but primary or elementary school goes beyond this period (6/7-10/12 year old children, depends on the countries’ school systems). 

We excluded only those studies, where the age range of children were too wide (from early childhood to teenagers)  and there were no separated data regarding 0-8 years old children.

  1. What do you define as empirical studies for eligibility and why only these?

A5: Thank you for your question. Our goal was to explore the research methods and results regarding the health literacy of young children. This is the reason, why we included studies if they were considered an empirical study (i.e., original data collection) and statistically tested. We did not aim to systematically analyze the concepts and the theoretical background.

  1. In one section you state you reduced sample from 238 to 12 and later under results you state you reduced sample from 513 to 12. Please explain this discrepancy.

A6: In the Methods section we write: ”We assessed 238 full articles (MCs, HBCs, JF), 18 of which proved to be relevant to the eligibility criteria of the study topic. Additional 6 articles were excluded due to specific reasons. Following the inclusion and exclusion criteria, 12 articles were included for the systematic review.” In the Results section: „After removing the duplications 513 research left. The 12 studies reviewed…”

After removing the duplications 513 research left, but after reading the abstract we need to exclude 275 studies. 238 full texts were read and after that only 12 were left. For better understanding, we rewrote this sentence in the Results Section.

  1. Once all the methodological details are provided, reviewers can assess the results that take up just one page here, and whether they are valid. Importantly, though, what is the “so what” of the findings? What is found that we did not already know – we already knew that oral health literacy, food, health care and health promotion might be prevalent topics, so what makes these topics unique for this age group. What uniquely and distinctly characterizes 12 articles that focus on young children?

A7: Thank you for your question. These 12 articles are important, because in the last 10 years only these focused on health literacy in early childhood and tried to measure the level of it directly among children. Other researches only draw conclusions from parents’ HL level to children’s. Yes, you are right these topics are prevalent, but when we summarize the results we can build up a program which can help develop not only these part of health but health literacy in general.  We explained it in detail in the Discussion section.

  1. In the discussion section, please provide the contributions of the findings to theory, to research, and to practice, and what makes the study compelling enough to publish.

A8: Thank you for your suggestion. We expanded the Discussion section with some other data. The included studies cannot give new information about the topics background, but the research methodology and the result related to health literacy are new findings. Summarizing these in a systematic review, we considered important in order to define new goals for the development of early childhood health.

We made some other corrections, clarifications and additions. Please find our corrections in the new uploaded manuscript and appendix.

We kindly request you to accept the manuscript for publication.

Yours sincerely, on behalf of the authors,

Henrietta BÁNFAI-CSONKA

subject teacher, PhD student

University of Pecs
Faculty of Health Sciences
H-7621 Pecs, Vorosmarty street 4.
Phone: +36/72/513-671
E-mail: csonka.henrietta@pte.hu

Round 2

Reviewer 1 Report

The authors have performed all suggested changes and the manuscript has been substantially improved

Reviewer 2 Report

I am very impressed with the significant revisions completed in the paper and the respect the authors provided to the reviewer's comments. This is a sound research paper and it has addressed all of the major concerns that I had.